# Genetic Diversity and Population Structure of Polish Konik Horse Based on Individuals from All the Male Founder Lines and Microsatellite Markers

**DOI:** 10.3390/ani10091569

**Published:** 2020-09-03

**Authors:** Agnieszka Fornal, Katarzyna Kowalska, Tomasz Zabek, Agata Piestrzynska-Kajtoch, Adrianna D. Musiał, Katarzyna Ropka-Molik

**Affiliations:** Department of Animal Molecular Biology, National Research Institute of Animal Production, 32-083 Krakow, Poland; katarzyna.kowalska@izoo.krakow.pl (K.K.); tomasz.zabek@izoo.krakow.pl (T.Z.); agata.kajtoch@izoo.krakow.pl (A.P.-K.); adarenazyga@interia.pl (A.D.M.); katarzyna.ropka@izoo.krakow.pl (K.R.-M.)

**Keywords:** population differentiation, parentage control, STR markers, horse, Polish Konik

## Abstract

**Simple Summary:**

We investigated the genetic diversity of Polish Konik in sire lines. The Polish Konik horse is a Polish native horse breed of a primitive type and is one of the breeds managed via a conservation program. Due to the small number of breed founders (six male lines) and paternal lineages with unknown representation, the genetic diversity and structure of the population should be monitored regularly. This is the first characterization of the genetic diversity of Polish Konik paternal lines based on microsatellite markers (STRs). The fixation index (F_ST_), which measures population differentiation, was low for all the studied markers and its mean value was 0.0305. Our analysis also revealed that there has been no inbreeding in the Polish Konik population for the studied group. This monitoring of genetic diversity could be helpful for the Polish Konik breeding and conservation programs and also for making informed decisions in the management of paternal lineages.

**Abstract:**

The Polish Konik horse is a primitive native breed included in the genetic resource conservation program in Poland. After World War II, intensive breeding work began, aimed at rebuilding this breed. Now, the whole Polish Konik population is represented by six male founder lines (Wicek, Myszak, Glejt I, Goraj, Chochlik and Liliput). Individuals representing all six paternal lineages were selected based on their breeding documentation. We performed a fragment analysis with 17 microsatellite markers (STRs) recommended by the International Society for Animal Genetics (ISAG). The genetic diversity and structure within the paternal lineages and the whole of the studied group were investigated. The average allelic richness was 6.497 for the whole studied group. The fixation index (F_ST_; measure of population differentiation) was low (about 3%), the mean inbreeding coefficient (F_IT_) was low and close to 0, and the mean inbreeding index value (F_IS_) was negative. The mean expected heterozygosity was established at 0.7046 and was lower than the observed heterozygosity. The power of discrimination and power of exclusion were 99.9999%. The cumulative parentage exclusion probability equaled 99.9269% when one parental genotype was known and 99.9996% with both parents’ genotypic information was available. About 3% of the genetic variation was caused by differences in the breed origin and about 97% was attributed to differences among individuals. Our analysis revealed that there has been no inbreeding in the Polish Konik breed for the studied population. The genetic diversity was high, and its parameters were similar to those calculated for native breeds from other countries reported in the literature. However, due to the small number of breed founders and paternal lineages with unknown representation, the population’s genetic diversity and structure should be monitored regularly.

## 1. Introduction

The Polish Konik horse (*Equus caballus*) is a Polish native horse breed of a primitive type and is one of the breeds managed via a conservation program [1]. There is a hypothesis that the Polish Konik horse exhibits the external features of the extinct Tarpan (*Equus ferus ferus*) and there is some evidence that Tarpans were crossbred with local primitive domestic horses [2]. Besides wild-type mouse-colored coats being a good adaptation to life in forest conditions [3], Polish Konik horses (Figure 1) present a characteristic primitive coat color—a Dun type with a predominant black Grullo color [4]. The name “Polish Konik” was introduced by Professor Tadeusz Vetulani in 1925, and to date “Polish Konik Horse” is the nomenclature recommended by the Polish Horse Breeders Association. The breeding history of the Polish Konik horses is nearly 100 years old and began in 1923, when the first individuals were placed in the oldest National Polish Stud in Janów Podlaski. The first Polish Konik Horses reserve was established in the Białowieża Forest in 1936. After World War II, the population of Polish Konik horses was rebuilt from a small number of surviving individuals [1,2,5,6,7,8]. Forming a stud farm in Popielno in 1954 allowed for the collection of all surviving individuals in one place. In 1955, the Polish register was released, and in 1962, the first volume of the studbook for the breed was established [9]. There are two ways of maintaining Polish Konik breeding. The first is a traditional system in stabled conditions, and the other is maintenance in a reservation—i.e., living in the wild. Initially, for the first 20 years of existence of the Polish Konik horse farm in Popielno, breeding work in the stable group was focused on adapting this type of Polish horse to the needs of agriculture at that time. Since the 1970s, as a part of the breeding program of the breed, selection in this group has been focused on preserving the breed’s characteristic features, including their typical behavior, color, good adaptation to the local environment, low feed requirements, healthiness, fertility, and high utility as carriage and saddle horses. In reserves, horses have mainly undergone natural selection for health, endurance in environmental conditions, low nutritional requirements, resourcefulness in obtaining food, fertility and the ability to reproduce by random mating, and appropriate social behavior that allows for the creation and maintenance of the herd [10].

Presently, the population consists of 34 maternal and 6 paternal lineages [11] with the closed studbook [12], but from the maternal lines, only 16 are active (the apparent progression for only 6 lineages) [13]. Since 1984, the Polish Konik horses have been deemed a pure breed, and the addition of the blood of other breeds is forbidden [14]. According to the breeding program, balance in the maintenance of all existing maternal and paternal lines is recommended to avoid the potential negative effects of inbreeding [11]. Among the male lines (Wicek, Goraj, Chochlik, Myszak, Glejt I, and Liliput), stallions from all of the paternal lines play active roles in breeding activities. According to Pasicka and also to Tomczyk-Wrona, the least numerous were the lines represented by Liliput and Glejt I [9,14]. According to the newest research presented by Jaworski et al. [15], nowadays, the least numerous are the lines represented by Myszak and Glejt I. The percentage share of male lines reported in 2019 in the population of Polish Konik was: Wicek—34.4%; Myszak—9.6%; Glejt I—9.6%; Goraj—17.2%; Chochlik—18.2%; Liliput—11% [15]. 

The maintenance of the primitive traits of this breed (inter alia, robustness to diseases or good fertility) is one of the most important goals of the Polish Konik conservation breeding program and is strictly adhered to. The conservation program of this breed is important for biodiversity preservation in post-agricultural areas, wastelands, and forest areas, and for landscape maintenance [2]. Primitive breeds of horses like the Polish Konik (as well as cattle) are especially suitable for grazing in valuable nature areas [16].

Due to maternal and paternal line analyses indicating that their uneven representation (according to Jaworski (Jaworski, 1997 in [17])) [17] and the relatively small population of Polish Konik horses, the genetic variability of this breed should be monitored, especially given that their present genealogical statistics are unknown. Polish Konik horses, which are unique, natural-breeding relicts, must be preserved as a specific reservoir of genetic resources for future breeding work [8]. Parentage control has been provided for all Polish Konik horses in Poland since 2009 [13] according to International Society for Animal Genetics (ISAG) recommendations [18].

Microsatellite markers are commonly used in structure and genetic diversity analyses (within-population and between populations) [1,19,20,21,22,23,24]. The genetic diversity and population structure of Polish Konik paternal lineages, studied via microsatellite markers (STRs), could give information about variability; the results could be compared with those of other horse breeds and could be a tool to monitor the breed. The aim of our study was to characterize the genetic diversity of Polish Konik paternal lines based on microsatellite markers.

## 2. Materials and Methods

We examined the genetic variability and structure of the Polish Konik breed among all six paternal lines. We used blood and hair follicles samples for routine horse parentage testing (via collection of DNA samples from different regions of Poland) and investigated the pedigree of the selected samples. The number of selected samples for each paternal line does not correspond with the paternal line’s size. We could not find any comprehensive reports about general population statistics. We eliminated all samples presenting a doubtful or excluded pedigree. Finally, we tested 196 Polish Konik samples representing the six paternal lineages. The distribution of the individuals among the paternal lineages was as follows: Wicek—48; Myszak—12; Glejt I—35; Goraj—32; Chochlik—36; Liliput—33. These numbers of the individuals as a percentage share of 196 samples are comparable to the percentage share of male lines reported in 2019 by Jaworski et al. [15].

The DNA samples were isolated from 300 μL of peripheral blood using a Wizard Genomic DNA Purification Kit (Promega, WI, USA) or Sherlock AX (A&A Biotechnology, Gdynia, Poland) and from hair follicles using Sherlock AX (A&A Biotechnology). DNA was amplified in a multiplex reaction (17 loci) using an Equine Genotypes Panel 1.1 Kit (ThermoFisher Scientific, Waltham, MA, USA) with positive and negative DNA controls. The amplification of STRs was performed according to the manufacturer’s protocols using 17 primer pairs labelled with four fluorescent dyes. We analyzed the PCR products in a capillary electrophoresis system using a 3130xl Genetic Analyzer (Applied Biosystems, Foster City, CA, USA). We conducted products separation using a GeneScan-500 LIZ Size Standard (Applied Biosystems), enabling automated DNA fragment analysis. Then, we processed the received data in GeneMapper 4.0 software (Applied Biosystems). This method was regularly checked and validated through ISAG horse parentage comparison tests and standardized following ISAG recommendations. The DNA profiles were specified using 12 markers from the ISAG core STR panel (AHT4, AHT5, ASB2, HMS2, HMS3, HMS6, HMS7, HTG10, HTG4, VHL20, ASB17, and ASB23) and 5 markers from the ISAG additional STR panel (HTG6, HTG7, CA425, HMS1 and LEX3).

We analyzed the genetic diversity of the six paternal populations via GenAlEx6 [25] and FSTAT v.2.9.3 [26] to estimate the F statistic.

We used Structure 2.3.4 software to analyze the genetic structure of the populations by Bayesian clustering methods [27]. Markov chain Monte Carlo (MCMC) parameters were set on a burn-in period of 50,000 and 150,000 iterations. We set the number of clusters (K) from 1 to 8 in the procedure, and at the next step we estimated ∆K using Structure Harvester v0.6.94 [28]. However, because of the low population size, the estimation is of a merely illustrative nature.

Due to a lack of differentiation among the paternal lineages and the small amount of six paternal populations, the following analyses were conducted for all samples as a whole population. We conducted population genetic analyses using the pegas package in R [29] and our own software (self-designed application counting the statistics described by Huston [30]). The observed and expected heterozygosity (H_o_ and H_e_) [31,32], mean inbreeding index value (F_IS_) [33], and polymorphism information content (PIC) [34] were estimated using the pegas package. We also estimated other coefficients using our self-made software based on a method suggested by Huston [30]: we calculated the power of discrimination (PD), combined probability of exclusion (PE), probability of exclusion for each locus for one of the known parental genotypes (PE_1_) and both parental genotypes (PE_2_) and the probability of identity (P_ID_).

## 3. Results

The results of the analyses of 17 microsatellite loci in six paternal populations, shown in Table 1, indicate that the paternal lineages are similar and, generally (similar mean number of alleles, frequency of alleles and heterozygosity), may be treated as a single population for this study. The overall N_a_ for all 196 samples was 6.157. We also evaluated Nei’s genetic distances and the highest differentiation was identified between the Liliput and Myszak (D_A_ = 0.120)

The results for the structure analyses (Structure 2.3.4 software), run to investigate the population structure and eventual admixture patterns among the six paternal populations, are shown in Figure 2 and Figure 3. The first figure displays the highest ΔK calculated using structure (MCMC iterations were done for each K value between 1 and 8, with K representing the putative clusters that could be recognized within the population). Visualizations of the population structures for ΔK = 2 and ΔK = 3 are presented in Figure 3.

Taking into consideration the analyzed data, ∆K = 2 and ∆K = 3 presented in Figure 1 are the most probable. This suggests that there could be two genetic clusters. It is caused by a low genetic distance among the six lineages, therefore we provided a further analysis for all samples without division into paternal lineages. The allelic richness computed for each locus in particular populations was estimated. The average allelic richness was 6.497 for all loci in all six lineages. The lowest allelic richness per locus (below 4.985) was observed at loci HTG4, HTG6, and HTG7 and the highest (from 8.1755 to 9.158)—for ASB2, HTG10, VHL20, and ASB17. The mean number of alleles for all samples was 7.176, and the mean effective number of alleles was 3.854.

The F-Statistics results of the whole studied population are presented in Table 2. The fixation index (F_ST_), which measures population differentiation, was low for all studied markers and its mean value was 0.0305. The mean F_IT_ was low and close to 0 (0.0189). The F_IS_ was negative (its mean value was −0.0134), which suggested that the selected samples in these analyses probably belong to unrelated individuals among the paternal lineages. The mean values were calculated without the X-chromosome linked LEX3 locus.

The population coefficients were calculated and are presented in Table 3. 

The mean expected heterozygosity was 0.7046 and was lower than the observed heterozygosity (0.6519). The power of discrimination and power of exclusion were both close to 1 (99.9999% and 99.9967%). The values of PIC, H_o_ and H_e_ coefficients (above 0.6) are very high values except HTG4, HTG6, HTG7, and LEX3. The cumulative power of discrimination and the probability of identity were high. Based on the probability of exclusion (PE_1_, PE_2_), we calculated the cumulative parentage exclusion probability knowing one (CPE_1_) or both (CPE_2_) parents. CPE_1_ amounted to 99.9269%, and CPE_2_ was 99.9996%.

## 4. Discussion

Features of Polish Konik that are recognized in their conservation program—such as hardiness, natural adaptive instincts, and potential for controlling dense forest invasion of natural wetland areas—have been noticed by environmentalists and scientists in Europe. Polish Konik horses were exported to the Netherlands and then selected to maintain their most favorable phenotypic characteristics. Now, they are called the Dutch Konik horse and—according to the studbook—are descendants of the Polish Konik horse breed [35]. Another interesting fact is that the Polish Konik horses had an influence on foreign horse breeding of wild horses from Dülmen in Westphalia (Germany) [9]. 

It has been proven that genetic diversity analysis based on Y chromosome variability can be an informative source of data to explain differentiation and the origin of the male founders of many horse breeds [36,37]. In turn, mitochondrial DNA provides information about female founders. To date, there is no recommended panel of SNPs for equine parentage testing. The ISAG panel of STRs is universally accepted and can provide information on population diversity and genetic structure. This is the first study to compare the genetic structure of Polish Konik paternal lines based on STR analysis. Our structure analyses showed the similarities of the paternal lineages. Moreover, due to the small numbers of individuals representing the selected paternal groups, we treated the six paternal populations as a single population. Bayesian clustering indicated that the number of genetic groups in our study (most probably, there are two genetic clusters) does not correspond with the number of six lineages. The F_ST_ level for all samples (the whole studied group) points to the small diversity of the analyzed loci, but the low value of this coefficient is probably due to the nonrandom mode of sample selection. The F_ST_, which measures population differentiation, was low for all studied markers and its mean value was 0.0305, and is caused by breed differences. The F_IT_ values are close to zero, except for LEX3, and indicate that inbreeding depression is not a thread in the present study. The F_IS_ is negative for most loci, and this confirms the low level of inbreeding among the animals selected from the six paternal lineages and random mating within Polish Konik horses. The F_IS_ value is similar to those reported for Thoroughbred [19] or Czech Haflinger horses [20] and could be interpreted as no heterozygosity reduction in the population.

The Polish Konik population was rebuilt from the small number of surviving individuals after World War II. Our results show a low level of inbreeding, probably due to the well-provided breed management and high genetic diversity of the breed founders. However, a planned investigation of the maternal lines’ genetic diversity should give a broader context for these results. 

Nevertheless, compared to other breeds, the allelic polymorphism was high in the core panel and slightly lower in the additional panel. Allelic polymorphism was high in the core panel (except in HTG4) and rather low in the additional panel (except in CA425 and HMS1). We detected the lowest number of alleles in HTG6, where only three allelic variants were present. The calculated heterozygosity was similar to the value obtained in this locus for the Polish Konik horse breed by Gralak et al. [1] and Mackowski et al. [21] for 12 microsatellite markers. We also obtained a similar mean observed heterozygosity (a little lower than the value reported by Mackowski et al. [21]), which could be explained by the smaller number of samples in our study. The mean expected heterozygosity in studies by Mackowski et al. [21] was similar to that obtained in our case. The lower value of the observed and expected heterozygosity for HTG4, HTG6, and HTG7 were also shown by Gralak et al. [1] and Mackowski et al. [21]. On the other hand, the allelic richness and mean number of observed alleles in the mentioned loci were lower in our study. The mean number of alleles in this study was lower than that in the Polish Konik population reported by Castaneda et al. [22] based on 15 STR loci, but similar to that of the Estonian Native Horse (a breed genetically similar to Polish Konik) reported by the same researchers. Our results should not be compared with the results by Castaneda et al. because we have only 14 common STRs (a lack of LEX33 in our data). Castaneda et al. [22] tested 53 samples but did not provide any information about the origin of samples—there was no information as to whether samples came from Poland or from a population exported from Poland (potentially isolated from the Polish population for years). The goal of our research was to estimate the Polish Konik horses kept and registered in Poland (and next, other breeds of pony or other small breeds held in genetic conservation but with the same set of STRs). The lack of information about the selection of samples and about whether sampled individuals are the progeny of Polish stallions makes it difficult to explain the different genetic diversity of the Polish Konik horses presented by Castaneda et al. [22]. The overall N_a_ obtained by those authors was 7.1333 for 53 individuals, in comparison to our N_a_ of 6.157 for 196 samples. Probably, the reason for their higher N_a_ could be related with the genetic variations of other maternal lines or non-Polish origin of samples. The PIC, H_o_ and H_e_ coefficients were also high (above 0.5) for all markers except HTG4 and HTG6 in Gralak et al. [1] and Fornal and Radko [23]. The overall H_o_ and H_e_ were similar to those reported by Castaneda et al. [22] for Polish Konik horses, but lower than those detected for Pottoka (a semiferal pony) or Jaca Navarra, two horse breeds in a conservation program in Spain, reported by by Solis et al. [24] and Rendo et al. [38]. In turn, the PIC is analogous to the value reported for the Terceira Pony, an Azorean horse breed held in genetic conservation [39]. High values indicate the most informative markers for describing population diversity. The high values obtained for the calculated coefficients point to the great effectiveness of the applied marker set for a population study in the investigated breed. The lowest informative locus, LEX3, should not be interpreted because it is X-chromosome linked, so it is not randomly selected. The next step in our investigation would be a comparison of the genetic diversity and population structure of the maternal lines, which would allow a comprehensive and intentional development of this breed with proper proportions across all existing maternal and paternal lines. The obtained data would be useful for a further assessment of genetic variation in Polish Konik horse breeding in Poland.

## 5. Conclusions

This study aimed to assess the genetic diversity and population structure of Polish Konik individuals selected by their paternal lines. Our study was conducted on the current STR set (17 loci) recommended by the ISAG and it is the first such research on Polish Konik as previous studies only included 12 STR markers. We performed the analysis with representatives of all paternal lineages, which probably allowed us to include all variants coming from sire lines. Despite the small number of paternal lineages in our research, the genetic diversity parameters were similar to those obtained for other native breeds in other countries reported in the literature. Surprisingly, the inbreeding index indicates that there is no inbreeding in the analyzed Polish Konik population, despite the fact that the population experienced a bottleneck event of genetic diversity several decades ago. Despite the lower parameters for the two loci, overall, the genetic variation of the STR markers was high and similar to that obtained in other research on this breed. However, the breed’s genetic structure should be monitored because the population was derived from a small number of individuals and particular paternal lineages now have an uneven representation. This monitoring could be helpful for the Polish Konik breeding and conservation program and also for making informed decisions in paternal lineages management. 

## Figures and Tables

**Figure 1 animals-10-01569-f001:**
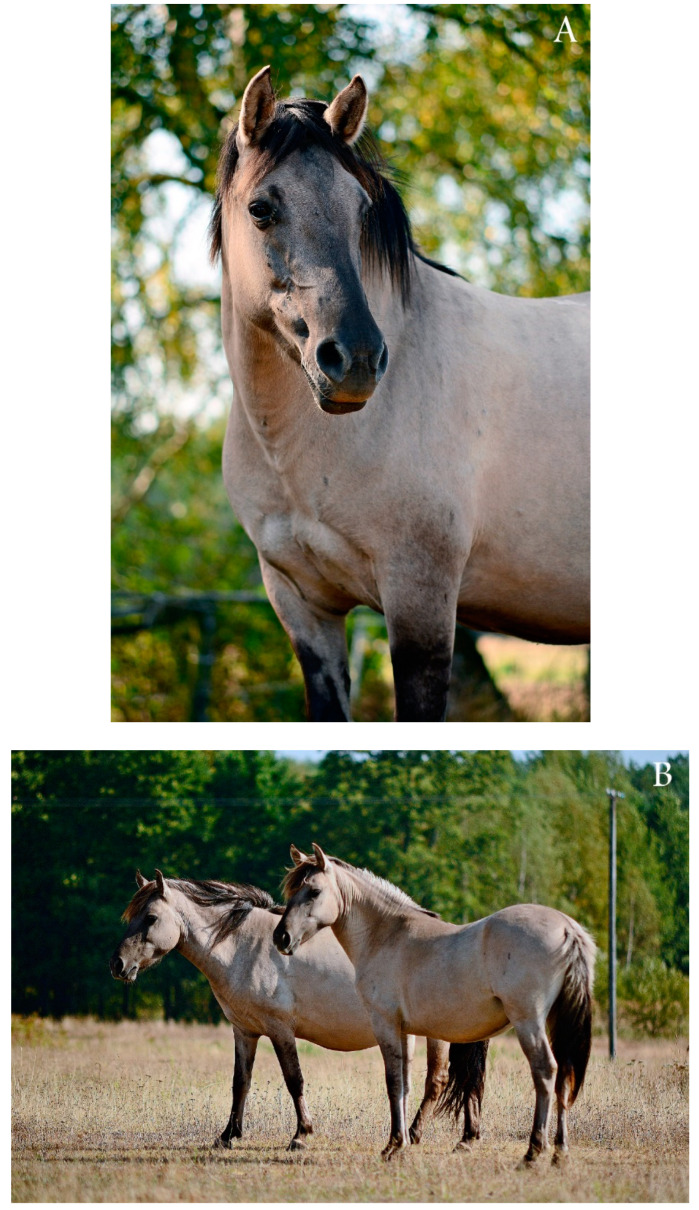
(**A**) Polish Konik; (**B**) appearance of Polish Konik (photo by Paulina Peckiel).

**Figure 2 animals-10-01569-f002:**
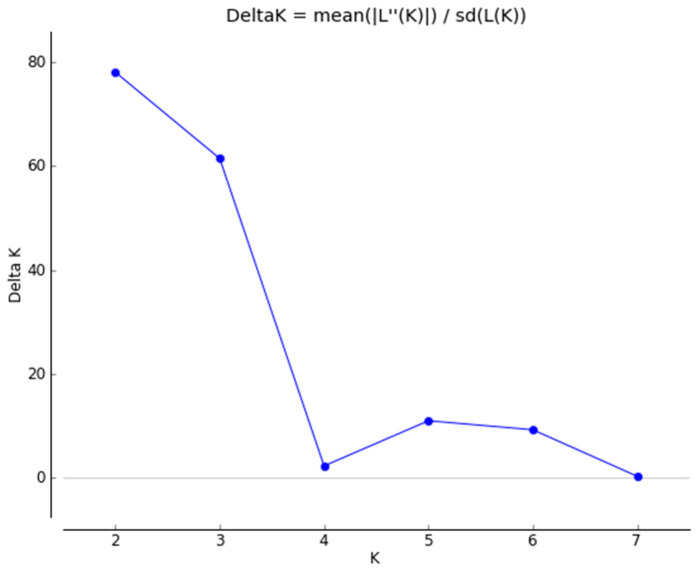
The highest likelihood and ∆K were observed for K = 2 and K = 3.

**Figure 3 animals-10-01569-f003:**
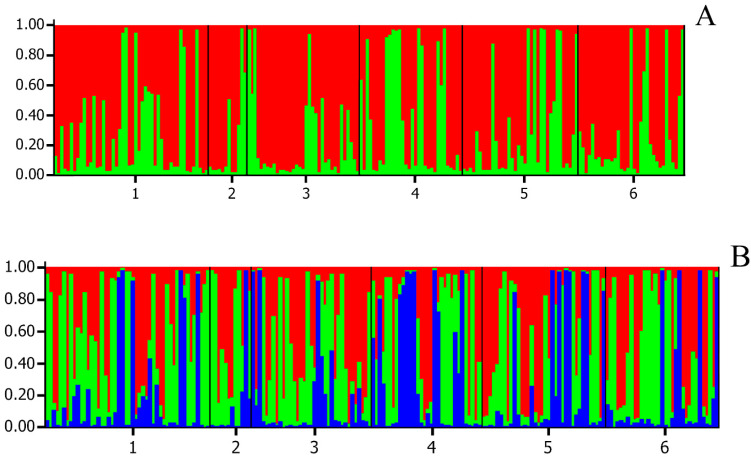
Structure analysis for K = 2 (**A**) and K = 3 (**B**) of six Polish Konik paternal lineages based on the ∆K method. 1, Wicek; 2, Myszak; 3, Glejt I; 4, Goraj; 5, Chochlik; 6, Liliput.

**Table 1 animals-10-01569-t001:** Characteristics of 17 microsatellite loci in six paternal lineages (*n* = 196).

Paternal Lineage	N	N_a_	N_a_ Freq. ≥ 5%	A_e_	I	No. Private Alleles	H_e_
Wicek	48	6.824	4.647	3.824	1.476	0.412	0.693
Myszak	12	5.059	4.000	3.306	1.302	0.000	0.657
Glejt I	35	6.294	4.588	3.875	1.453	0.059	0.700
Goraj	32	6.118	4.176	3.627	1.396	0.000	0.678
Chochlik	36	6.471	4.765	3.730	1.449	0.000	0.689
Liliput	33	6.176	4.588	3.715	1.429	0.059	0.688

Mean values per paternal lineages: N_a_—mean number of alleles; N_a_ freq. ≥ 5%—mean number of alleles for which the frequency is equal to or lower than 5%; A_e_—no. of effective alleles; I—information index; No. of private alleles—mean number of private alleles across the population; H_e_—expected heterozygosity.

**Table 2 animals-10-01569-t002:** Genetic structure coefficient based on F-statistics: inbreeding coefficient (F_IT_), coefficient of differentiation (F_ST_) and mean inbreeding index value (F_IS_) for each locus.

	F_IT_	F_ST_	F_IS_
**AHT4**	−0.0096	0.0213	−0.0316
**AHT5**	0.0162	0.0373	−0.0219
**ASB2**	−0.0100	0.0113	−0.0215
**HMS2**	−0.0191	0.0053	−0.0245
**HMS3**	0.0231	0.0357	−0.0130
**HMS6**	−0.0185	0.0269	−0.0466
**HMS7**	0.0613	0.0673	−0.0065
**HTG10**	−0.0061	0.0211	−0.0278
**HTG4**	0.0342	0.0099	0.0246
**HTG6**	0.0000	0.0526	−0.0555
**HTG7**	0.0105	0.0244	−0.0143
**VHL20**	0.0261	0.0438	−0.0185
**ASB17**	0.0142	0.0166	−0.0024
**ASB23**	0.0449	0.0262	0.0193
**CA425**	0.1052	0.0756	0.0319
**HMS1**	0.0305	0.0362	−0.0059
**LEX3**	0.9562	0.0079	0.9559
**Mean**	0.0741	0.0305	−0.0134

F_IT_, inbreeding coefficient of an individual relative to the total population; F_ST_, coefficient of differentiation, fixation index; F_IS_, deviation from Hardy-Weinberg proportions (within-population inbreeding coefficient).

**Table 3 animals-10-01569-t003:** Population coefficients for all loci in the 17 microsatellite markers (STR) set.

	H_o_	H_e_	PD	PE	PIC	PE_1_	PE_2_	P_ID_
**AHT4**	0.7602	0.7482	0.8914	0.5274	0.7088	0.3486	0.5261	0.1028
**AHT5**	0.7908	0.7964	0.9280	0.5821	0.7672	0.4240	0.6020	0.0706
**ASB2**	0.8010	0.7895	0.9303	0.6009	0.7670	0.4333	0.6129	0.0669
**HMS2**	0.7398	0.7234	0.8819	0.4925	0.6857	0.3207	0.5013	0.1142
**HMS3**	0.7653	0.7763	0.9101	0.5363	0.7431	0.3902	0.5694	0.0832
**HMS6**	0.8163	0.7956	0.9222	0.6297	0.7638	0.4146	0.5929	0.0735
**HMS7**	0.6684	0.7015	0.8654	0.3810	0.6519	0.2914	0.4610	0.1387
**HTG10**	0.8112	0.8012	0.9326	0.6200	0.7747	0.4393	0.6160	0.0661
**HTG4**	0.4541	0.4681	0.6885	0.1504	0.4421	0.1198	0.2785	0.3090
**HTG6**	0.1684	0.1663	0.3011	0.0217	0.1593	0.0138	0.0843	0.7020
**HTG7**	0.6173	0.6195	0.7915	0.3122	0.5522	0.1994	0.3475	0.2121
**VHL20**	0.7806	0.7931	0.9267	0.5636	0.7683	0.4344	0.6121	0.0676
**ASB17**	0.7857	0.7926	0.9197	0.5728	0.7662	0.4311	0.6081	0.0694
**ASB23**	0.7092	0.7371	0.8827	0.4426	0.6954	0.3333	0.5094	0.1108
**CA425**	0.6480	0.7122	0.8793	0.3524	0.6813	0.3198	0.5046	0.1137
**HMS1**	0.7296	0.7457	0.8956	0.4755	0.7124	0.3541	0.5358	0.0979
**LEX3**	0.0357	0.8110	0.8230	0.0012	0.7858	0.4552	0.6307	0.0609
	0.6519 *	0.7046 *	0.9999 **	0.9999 **	0.6721 *	0.9992 **	0.9999 **	5.04 × 10^−17^ **

H_o_, H_e,_ observed and expected heterozygosities; PD, the power of discrimination; PE, the combined probability of exclusion; PIC, polymorphic information content; PE_1_, PE_2_, the probability of exclusion for each locus for one known parental genotypes and for both parental genotypes, respectively; P_ID_, the probability of identity; * mean value; ** cumulative value.

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
