# Peer review of "Genetic Diversity and Population Structure of Polish Konik Horse Based on Individuals from All the Male Founder Lines and Microsatellite Markers"

_animals, 2020, doi:10.3390/ani10091569_

Round 1

Reviewer 1 Report

I have reviewed the revised manuscript. There are some major problems which the authors should address.

  1. The content

(1) In Line 211, the authors stated that 3% of the genetic variations were caused by the sire lines and the rest 97% was attributed to the genetic difference of individuals in the studied population. They said that the results were drawn from the value of Fst (in F-statistics). However, Fst is usually used to estimate the genetic distance among sub-populations in total population. As a parameter derived from heterozygosity of total population and sub-populations, Fst is not a proper value to determine the contribution of genetic variation of populations. What is more, value of Fst varies from 0 to 1, so please do not describe it as a percentage in Line 32.

(2) from Line 61 to 62, the authors stated that they merged samples from the five paternal lines because there are only two genetic clusters revealed by the analysis of STRUCTURE. However, this is not a good reason to do so. The very low genetic distance among the five paternal lines is a convincing reason to merge the data.

(3) The discussion part is not well-organized It should be improved and re-organized. It is suggested that the authors should refer to previous published reports in this field.

(4) Although authors explained that the detailed information of samples was not available to them, the authors should be aware of that their samples are lack of some key information, which may make their results less convincing. In formal reports on genetic diversity, the distribution of the samples is always documented.

  1. The English writing of the manuscript

There are too many writing mistakes in the manuscript, For example:

Line 19, "there is was no inbreeding". It is obviously that "is" and "was" could not be used together.

Line 15 and 42, " lineages unknown representation". Is that "lineages with unknown representation"?

Line 77, "According Pasicka", should be corrected to "according to Pasicka".

……….

Though I understand that lot of endeavors from the authors were involved in the genotyping and data analysis, the current status of the manuscript could not meet that basic standards of publication. More efforts should be made to improve their manuscript.

Author Response

Review 1 Report (Round 1)

  1. The content

(1) In Line 211, the authors stated that 3% of the genetic variations were caused by the sire lines and the rest 97% was attributed to the genetic difference of individuals in the studied population. They said that the results were drawn from the value of Fst (in F-statistics). However, Fst is usually used to estimate the genetic distance among sub-populations in total population. As a parameter derived from heterozygosity of total population and sub-populations, Fst is not a proper value to determine the contribution of genetic variation of populations. What is more, value of Fst varies from 0 to 1, so please do not describe it as a percentage in Line 32.

We corrected this sentence: “was caused by breed differences”.

And also we corrected Line 38: “by differences in breed”

Yes, Fst varies from 0 to 1, but some authors presenting Fst as percentage value. We think % presentation of value will be easier to understand by readers who are not an expert.

Fst preseted as a value from 0 to 1 is common, but as a percentage is also using . Examples of papers where authors describe Fst as a percentage:

Canon, J., Checa, M. L., Carleos, C., Vega‐Pla, J. L., Vallejo, M., & Dunner, S. (2000). The genetic structure of Spanish Celtic horse breeds inferred from microsatellite data. Animal genetics, 31(1), 39-48.

Geng, Y., Yang, Z. P., Chang, H., MAO, Y. J., SUN, W., GUO, X. Y., & QU, D. Y. (2007). Genetic differentiation and gene flow among six breeds of Mongolian group in China. JOURNAL-YANGZHOU UNIVERSITY AGRICULTURAL AND LIFE SCIENCES EDITION, 28(3), 22.

Hoda, A., Sena, L., & Hykaj, G. (2012). Genetic diversity revealed by AFLP markers in Albanian goat breeds. Archives of Biological Sciences, 64(2), 799-807.

Georgescu, S. E., Manea, M. A., Zaulet, M., & Costache, M. (2009). Genetic diversity among Romanian cattle breeds with a special focus on the Romanian Grey Steppe Breed. Romanian Biotechnological Letters, 14(1), 4194-4200.

2) from Line 61 to 62, the authors stated that they merged samples from the five paternal lines because there are only two genetic clusters revealed by the analysis of STRUCTURE. However, this is not a good reason to do so. The very low genetic distance among the five paternal lines is a convincing reason to merge the data.

 We corrected Line 161-162:

“It suggested that there could be two genetic clusters,. It is caused by low genetic distance among six lineages, therefore we provided further analysis for all samples” […]

(3) The discussion part is not well-organized It should be improved and re-organized. It is suggested that the authors should refer to previous published reports in this field.

It is hard to discuss all our results with other breeds in systematic way. Our major goal of the discussion was comparing with results for Polish Konik breed obtained by other researchers. Heterozygosity were discussed in Discussion where we compare our data with literature data for this breed. Other coefficient where discussed for the same breed if the data were available, or with other breeds – when the data for Polish Konik wasn’t available. We tried to compare the data with other breeds, but we have chosen breeds like pony or other small breeds held in genetic conservation due to their similarity to Polish Konik (environment, type etc.)

 (4) Although authors explained that the detailed information of samples was not available to them, the authors should be aware of that their samples are lack of some key information, which may make their results less convincing. In formal reports on genetic diversity, the distribution of the samples is always documented.

We add the data in “Introduction” and “Materials and Methods”.

  1. The English writing of the manuscript

There are too many writing mistakes in the manuscript, For example:

Line 19, "there is was no inbreeding". It is obviously that "is" and "was" could not be used together.

Line 15 and 42, " lineages unknown representation". Is that "lineages with unknown representation"?

 Line 77, "According Pasicka", should be corrected to "according to Pasicka".

We corrected

We add corrections English-Editing-Certificate-20705 into manuscript.

Reviewer 2 Report

The manuscript is much improved in detail, but my basic criticism remains the same: it makes little sense to describe a panmictically maintained captive population with F_st, F_is, and F_it. As is obvious documented, the paternal lines are not used in breeding to substructure the population; expectedly this is a result of the author's genetic analysis. Even if a population would be highly inbred but panmictic and arbitrarily divided (eg by male founder lines), F_st, F_is, and F_it would be nearly zero. For meaningful comparisons, more horse populations should be added and then F_st, F_is, and F_it should be re-calculated to allow for the conclusions of the authors.

I also add a commented version of the manuscript for the authors.

Author Response

Review 2 Report (Round 1) 

The manuscript is much improved in detail, but my basic criticism remains the same: it makes little sense to describe a panmictically maintained captive population with F_st, F_is, and F_it. As is obvious documented, the paternal lines are not used in breeding to substructure the population; expectedly this is a result of the author's genetic analysis. Even if a population would be highly inbred but panmictic and arbitrarily divided (eg by male founder lines), F_st, F_is, and F_it would be nearly zero. For meaningful comparisons, more horse populations should be added and then F_st, F_is, and F_it should be re-calculated to allow for the conclusions of the authors.

Data set are representing whole country population of Polish Konik by their male lines. Because our data set representing whole country population condition of being population panmictic is not met. Also population panmictic criteria are not met because of geographical distance of whole country population, behavioural of Polish Konik, sealing animal, and so on.

Moreover, we shoved data set, manuscript and result to evolutionary biologist who know this breed and that person confirm, that is not a panmictic population.

I also add a commented version of the manuscript for the authors.

Line 48 “one of the breeds” – Polish Konik is one from seven breeds managed with conservation program in Poland, that’s why we wrote “one of”

Line 48 – It is hypothesis, so we think it should stay that form.

Line 50, 53, 57 – corrected.

Line 58 – we mean that Popielno as a stud was organized/formed.

Line 67  – corrected

Line 73 – it was deemed as - acclaim as, as a formal start point of maintenance as pure

Line 82 – We agree, but we think this sentence is needed for eventual readers who knows this area not much.

Line 90 comment – Yes, assertion of paternity.

Line 161 comment – Yes, we agree. But we tested few times with little different parameters of simulations of this probability. Always we have get such results.

Line 169 comment – Correct, populations were defined as paternal lines.

Line 176 comment – comment is empty

Line 186 comment – Power of discrimination is calculated based on matching probability,  power of exclusion is defined as the fraction of individuals having a DNA profile that is different from that of a randomly selected individual in a typical paternity case. Describe in line 135-138.

Line 203 comment – We add: “Final and recommended panel of SNPs for parentage was not release for this moment. So, STRs” […]

Line 205 comment – comment is empty

Line 209 comment – “FST level for all samples (the whole studied group)” – we mean, as in previous sentence – whole groups in our study of six lineages.

Line 210 comment – probably if we would take random sample form maternal and paternal lines, Fst could be higher.

Line 213 comment – corrected.

Line 224 comment – corrected.

Calculated heterozygosity was similar to heterozygosity for Polish Konik horse breed obtained by Gralak et al. [1] and Mackowski et al. [33] for 12 microsatellite markers.

Line 250 comment –

We didn’t have idea how write it better. English editor accept it in that form.

We add corrections English-Editing-Certificate-20705 into manuscript.

Reviewer 3 Report

The manuscript became substantially better. I have just a few comments.

1) Simple Summary, Line 19: Keep only “is” or “was”.

2) Abstract, Line 39: Take away “population” after Polish Konik or replace with “breed”

3) Materials and Methods, Line 106: Take away “)” after “33”.

4) Results, Line 141: Do you want to keep “or” between “alleles” and “heterozygosity” or replace “or” by “and”?

5) Results, Line 143: Move “Table 1. Characteristics…” to next page keeping the Table name and the table itself on the same page.

6) Results, Line 168: Remove a small black dot before “The F-Statistics …”.

7) Results, Lines 175 – 178, Table 2: I would recommend locating the entire Table 2 on the same page.

8) Results, Line 185: Replace a comma by a period in 0.7046.

9) Discussion, Line 209: Correct “linages” to “lineages”. Do you want to keep “(six)” after “lineages” or change to “number of six lineages”?

10) Discussion, Line 234: Missed [34] after “Castaneda et al.”

11) Discussion, Line 238: I recommend replacing “since” by “for”.

12) Discussion, Line 242: Missed [34] after “Castaneda et al.”

13) Discussion, Line 243: “our” instead of “ours” if you want to keep “Na”.

14) Discussion, Line 244: Replace “female lines other variants” by “other variants of female lines”.

15) Discussion, Line 244: “not-Polish” or “not Polish”?

16) Discussion, Line 250: I recommend deleting “for markers” after “High values …”.

17) Discussion, Line 253: Take away “of” after “because”.

18) References, Lines 307,  325-326, 327-329, 334-335, 339-340, 347-349, 360-363: Review that your references 10, 18, 19, 22, 25, 29, 34 are consistent with the reference guidelines and other references (lack of volumes, pages; capital or small letters; etc.).

Author Response

Review 3 Report (Round 1) 

1) Simple Summary, Line 19: Keep only “is” or “was”.

We corrected.

2) Abstract, Line 39: Take away “population” after Polish Konik or replace with “breed”

We corrected.

3) Materials and Methods, Line 106: Take away “)” after “33”.

We corrected.

4) Results, Line 141: Do you want to keep “or” between “alleles” and “heterozygosity” or replace “or” by “and”?

We corrected. “and”

5) Results, Line 143: Move “Table 1. Characteristics…” to next page keeping the Table name and the table itself on the same page.

We will check it in proofreading.

6) Results, Line 168: Remove a small black dot before “The F-Statistics …”.

We corrected.

7) Results, Lines 175 – 178, Table 2: I would recommend locating the entire Table 2 on the same page.

We will check it in proofreading.

8) Results, Line 185: Replace a comma by a period in 0.7046.

Corrected.

9) Discussion, Line 209: Correct “linages” to “lineages”. Do you want to keep “(six)” after “lineages” or change to “number of six lineages”?

number of six lineages

10) Discussion, Line 234: Missed [34] after “Castaneda et al.”

Corrected.

11) Discussion, Line 238: I recommend replacing “since” by “for”.

Corrected.

12) Discussion, Line 242: Missed [34] after “Castaneda et al.”

Corrected.

13) Discussion, Line 243: “our” instead of “ours” if you want to keep “Na”.

Corrected.

14) Discussion, Line 244: Replace “female lines other variants” by “other variants of female lines”.

Corrected.

15) Discussion, Line 244: “not-Polish” or “not Polish”?

“not Polish”

16) Discussion, Line 250: I recommend deleting “for markers” after “High values …”.

Corrected.

17) Discussion, Line 253: Take away “of” after “because”.

Corrected.

18) References, Lines 307,  325-326, 327-329, 334-335, 339-340, 347-349, 360-363: Review that your references 10, 18, 19, 22, 25, 29, 34 are consistent with the reference guidelines and other references (lack of volumes, pages; capital or small letters; etc.).

Corrected.

We add corrections English-Editing-Certificate-20705 into manuscript.

Reviewer 4 Report

The question the study sets out to answer is valid but a more holistic approach using both the pedigree and microsatellite data would provide more accurate insight.

You set out to conduct your analysis on the Polish population of the Konik horse to the exclusion of other animals that have been exported. What is the validity of this. Management of rare breeds by dividing up populations globally and keeping them genetically separate is a valid and accepted method but that then preculudes any imports or exports of animals or germinal products.

Much better to manage the global population (cf Lacey Ballou Ivy et al)

Other studies using the ISAG panel of markers have reported inbreeding in the Konik horse circa 7%. Your analysis seems to coroborate this with Fit across alleles of 0.0741 all be it this may be skewed by high homozygosity in Lex3.

The Konik is a modern recreation of an ancient breed that has a pedigree only going back circa 70 years. Given an average generation interval of 10 years in most equines that is 7 generations. The studbook is closed and there is no introgression from outside so it is inevitable that there will be an accumulation of inbreeding and accumulation at 1% per generation would not be out of the normal in a rare breed.

Microsatellite STR may be useful for parentage verification but assessment of genomic inbreeding is more appropriate using SNP's and Roh (run s of homozygosity). In your study you utilise the standard panel of 17 markers recognised by ISAG. The validity of any conclusions drawn from your study when compared to those that would come from a full genomic assessment of the 2.7 billion base pairs that make up the equine genome and identification of short and long runs of homozygosity raise some note of caution to anyone reading your paper and using your conclusions to shape breeding advice.

A more holistic approach would be to integrate molecular and pedigree based methods to help formulate breeding plans.

You clearly have access to studbook data to be able to select your samples from the 6 recognised sire lines.

Analysis with PopRep   (Groeneveld et al) using their webservice here:

https://popreport.fli.de/cgi-bin/entry.pl

would highlight levels of inbreeding; unbalanced parental contributions and Effective Population Size which should be the real drivers behind any breed management programme .

Author Response

Review 4 Report (Round 1) 

The question the study sets out to answer is valid but a more holistic approach using both the pedigree and microsatellite data would provide more accurate insight.

We agree, but we didn’t decide to use pedigree data in this paper. We agree that would be more holistic. But pedigree data could be too broad, rather it should be the topic of separate paper.

Moreover, is a competence of other research group.

You set out to conduct your analysis on the Polish population of the Konik horse to the exclusion of other animals that have been exported. What is the validity of this. Management of rare breeds by dividing up populations globally and keeping them genetically separate is a valid and accepted method but that then preculudes any imports or exports of animals or germinal products.

Much better to manage the global population (cf Lacey Ballou Ivy et al)

We don’t have access to data, and more importantly, samples from other countries where Konik is present. We have no knowledge if there is some international association (like for Hucul for example) or global data base of Konik.

Other studies using the ISAG panel of markers have reported inbreeding in the Konik horse circa 7%. Your analysis seems to coroborate this with Fit across alleles of 0.0741 all be it this may be skewed by high homozygosity in Lex3.

The Konik is a modern recreation of an ancient breed that has a pedigree only going back circa 70 years. Given an average generation interval of 10 years in most equines that is 7 generations. The studbook is closed and there is no introgression from outside so it is inevitable that there will be an accumulation of inbreeding and accumulation at 1% per generation would not be out of the normal in a rare breed.

Thank you for comment.

Microsatellite STR may be useful for parentage verification but assessment of genomic inbreeding is more appropriate using SNP's and Roh (run s of homozygosity). In your study you utilise the standard panel of 17 markers recognised by ISAG. The validity of any conclusions drawn from your study when compared to those that would come from a full genomic assessment of the 2.7 billion base pairs that make up the equine genome and identification of short and long runs of homozygosity raise some note of caution to anyone reading your paper and using your conclusions to shape breeding advice.

We agree. Bus SNPs are not commonly used in parentage testing in horses so far. STRs are much more universal to comparing with results of other populations at this moment. We add some information about SNPs in Discussion

A more holistic approach would be to integrate molecular and pedigree based methods to help formulate breeding plans.

You clearly have access to studbook data to be able to select your samples from the 6 recognised sire lines.

Analysis with PopRep   (Groeneveld et al) using their webservice here:

https://popreport.fli.de/cgi-bin/entry.pl

would highlight levels of inbreeding; unbalanced parental contributions and Effective Population Size which should be the real drivers behind any breed management programme .

As we mention is a competence of other research group.

We add corrections English-Editing-Certificate-20705 into manuscript.

Round 2

Reviewer 1 Report

The authors have improved their manuscript. However, following points should be addressed by them.

  1. The content

(1) The explanation of Fst

Fst is used to estimate the genetic distance among sub-populations in total population. As a parameter derived from heterozygosity of total population and sub-populations, Fst is not a value to estimate the contribution of genetic variation of populations.

This point was also mentioned in my comments of last review, but the authors did not correct it. As the statement may mislead the readers, the authors should make necessary correction and delete the sentences (Line 17-18; Line 221-222).

Although there are some reports in which Fst was described as percentages, it is not right for the authors to simply follow the previous publications without their own discrimination. The authors should follow the correct form, by which decimal should be used to represent Fst (Line 17-18; Line 221-222).

(2) The title of the manuscript

The title indicates that the content is mainly about the paternal lineages of polish Konik. However, the authors merely collected samples according to the paternal lineages, and did not conduct any analysis on the male horses of the paternal lines (Typical analysis on paternal lines were described in the studies on Y-chromosomal genetic markers (For examples, Felkel S et al, Sci Rep. 2019; Wallner et al, Curr Biol. 2017; Kreutzmann N, Anim Genet., 2014.)).

To avoid any misunderstanding, the authors should title their manuscript as "Genetic diversity and population structure of Polish Konik revealed with samples from different paternal lines and STR genotyping" or another title like that (Line 2-4).

(3) The reason of merging the samples from the five paternal lines

In Line 168-169, the authors stated that there is close genetic distance judged from the analysis of structure. But result displayed in Figure 2 mainly shows the genetic structures of the studied five sub-populations. Though the sub-populations seem to have similar genetic structures based on the analysis, the results of genetic distance among the five sub-populations should be showed to provide direct evidence which supports the authors' strategy of data analysis (the authors mentioned previously that they conducted the analysis of the genetic distances when they replied my comments of the first round. They may use the results they already have).

(4) The sample information

In Line 105, the authors stated that the samples are from different regions of Poland. The areas where the horses (or samples) are distributed and numbers of samples which were sampled in the areas should be provided.

(4) The order of presenting the results

The order that the authors presented their results is confusing. I suggest that the authors should present their results in following order: 1) The diversity of the 17 microsatellite loci; 2) Characteristics of 17 microsatellite loci in six paternal lineages; 3) analysis on the genetic structure of the populations; 4) analysis on the genetic distance of the five sub-populations; 5) F-statistics.

  1. Writing

In Line 47, "......and is one of the breeds......" should be changed to "......and one of the breeds......".

In Line 49-50, Change " Besides wild-type mouse-coloured giving......" to " Besides wild-type mouse-coloured coat giving......"

In Line 54, change " .....and begins in 1923" to " .....and began in 1923"

In Line 57, change "......the small number of ......" to "......a small number of ......"

In Line 63, change "......thise type of Polish horse......" to "......this type of Polish horse......"

In 77, change "....... all the lines' representatives demonstrate breeding acitvity" to " stallions from all of the paternal lines play active roles in breeding activities."

In Line 77-82, Please only keep the latest report on the population sizes of paternal lines, and the previous survey of Pasicka and Tomczyk-Wrona should be deleted. Please add a comma before " nowadays".

In Line 147, change " ......may be treated ......" to " ......and may be treated ......". Again, the authors should provide the results of analysis on genetic distance to support that all of the samples could be merged and analyzed together.

In Line 154-156, the methods should be assigned into Materials and Methods part.

In Line 169-170, the point" it is caused by low genetic distance among six lineages" should be supported by the results of analysis on genetic distance, as mentioned above.

In Line 173 change " .....was calculated for....." to " .....was observed at loci....."

In Line 174-175, the authors stated that the mean number of alleles for all samples was 7.765, and the mean effective number of alleles was 3.854. However, according to the results showed in Table 1, the number of alleles and the mean effective number of alleles calculated for each sub-population (paternal line) are less than the mentioned values (7.765 and 3.854). Given the very low private alleles of each sub-population, how could the authors get the mean values that are greater than all of the individual ones? Please check that carefully.

In Line 176, change "The F-Statistics results determined for the whole studied group’s STR markers……" to "The F-Statistics results of the whole studied population……"

In Line 183, change " ……for each locus in the data" to " ……for each locus "

In Line 193, Change "The mean expected heterozygosity was established as 0, .7046 and was lower than the observed heterozygosity (0.6519)" to "The mean expected heterozygosity was 0.7046 and lower than the observed heterozygosity (0.6519)".

In Line 195-196, Change "The PIC, Ho and He coefficients (above 0.6) had in general very high values for all but four loci (HTG4, HTG6, HTG7, and LEX3)" to "The values of PIC, Ho and He coefficients (above 0.6) are very high values except HTG4, HTG6, HTG7, and LEX3."

In Line 228, "influence" should be changed to "influences".

In Line 243, " but similar that in to Estonian population" should be replaced with "but similar with that of Estonian population"

In Line 255, please change" could be related with other female lines variants...." to " could be related with the genetic variations of other maternal lines...."

In Line 267, "......the maternal lines; this would ......" should be changed to "......the maternal lines, which would ......"

In 279-280", "....despite the small number of animals" should be changed to " .... Despite that the population experienced a bottleneck event of genetic diversity several decades ago."

Author Response

The authors have improved their manuscript. However, following points should be addressed by them.

Overall comment: thank you for all comments and suggestion. We tried to answer and include all, but our version of manuscript but we get manuscript annotated another numbers of lines than you pointed. I had to looking for the meaning to find directly which line do you mean. Please keep that in mind, we tried very hard not to miss anything and to improve manuscript according to your suggestion.

  1. The content

(1) The explanation of Fst

Fst is used to estimate the genetic distance among sub-populations in total population. As a parameter derived from heterozygosity of total population and sub-populations, Fst is not a value to estimate the contribution of genetic variation of populations.

This point was also mentioned in my comments of last review, but the authors did not correct it. As the statement may mislead the readers, the authors should make necessary correction and delete the sentences (Line 17-18; Line 221-222).

Although there are some reports in which Fst was described as percentages, it is not right for the authors to simply follow the previous publications without their own discrimination. The authors should follow the correct form, by which decimal should be used to represent Fst (Line 17-18; Line 221-222).

Thank you for comments. We didn’t find in handbooks proofs that Fst as percentages is incorrect form. Thus why we leaved percentages form.

Nevertheless we corrected Line 17-18 and:

We exchanged sentence into “The fixation index (FST), which measures population differentiation, was low for all studied markers and its mean value was 0.0305.”

And line 233-234 (I’m guessing that you mean that line, when you writing “Line 221-222”) we replaced with:

“FST, which measures population differentiation, was low for all studied markers and its mean value was 0.0305, it is caused by breed differences”

(2) The title of the manuscript

The title indicates that the content is mainly about the paternal lineages of polish Konik. However, the authors merely collected samples according to the paternal lineages, and did not conduct any analysis on the male horses of the paternal lines (Typical analysis on paternal lines were described in the studies on Y-chromosomal genetic markers (For examples, Felkel S et al, Sci Rep. 2019; Wallner et al, Curr Biol. 2017; Kreutzmann N, Anim Genet., 2014.)).

To avoid any misunderstanding, the authors should title their manuscript as "Genetic diversity and population structure of Polish Konik revealed with samples from different paternal lines and STR genotyping" or another title like that (Line 2-4).

We tested correctness and consistency of pedigree documentation until founders found. So we confirmed that we have all lines reported in literature and studbooks. So we are sure, that we have all lines.

All samples were have parentage testing verification since 2007, so we have few generation with confirmation that progeny of those founders are correct.

We are making analysis on the male horses of the paternal lines with Y-chomosome but other research (in progress).

So we made and still making holistic analyses of paternal line. So putting into title “from different paternal lines” could be confusing and suggesting that we took some lines, not all.

(3) The reason of merging the samples from the five paternal lines

In Line 168-169, the authors stated that there is close genetic distance judged from the analysis of structure. But result displayed in Figure 2 mainly shows the genetic structures of the studied five sub-populations. Though the sub-populations seem to have similar genetic structures based on the analysis, the results of genetic distance among the five sub-populations should be showed to provide direct evidence which supports the authors' strategy of data analysis (the authors mentioned previously that they conducted the analysis of the genetic distances when they replied my comments of the first round. They may use the results they already have).

In Lines 168-169 we have figure 1, so we are guessing that you describing old version of manuscript.

We add sentence in Line 156.

(4) The sample information

In Line 105, the authors stated that the samples are from different regions of Poland. The areas where the horses (or samples) are distributed and numbers of samples which were sampled in the areas should be provided.

We are guessing you mean Line 110. As we wrote in first round, the location of the samples is irrelevant – samples are from routine horse parentage testing, they wasn’t collected in the same time and during collecting samples the animals could be sold to another region of Poland, so in our opinion information about localisation are irrelevant.

(4) The order of presenting the results

The order that the authors presented their results is confusing. I suggest that the authors should present their results in following order: 1) The diversity of the 17 microsatellite loci; 2) Characteristics of 17 microsatellite loci in six paternal lineages; 3) analysis on the genetic structure of the populations; 4) analysis on the genetic distance of the five sub-populations; 5) F-statistics.

We have describe our workflow in Materials and Methods (and it was related with hypothesis that there is small population and it won’t be differences among six lineages).

Writing

In Line 47, "......and is one of the breeds......" should be changed to "......and one of the breeds......".

“is” was added by English editor

In Line 49-50, Change " Besides wild-type mouse-coloured giving......" to " Besides wild-type mouse-coloured coat giving......"

Line 52 corrected

In Line 54, change " .....and begins in 1923" to " .....and began in 1923"

Line 57 corrected, but English editor didn’t pointed to correction

In Line 57, change "......the small number of ......" to "......a small number of ......"

Line 59 corrected, but English editor didn’t pointed to correction

In Line 63, change "......thise type of Polish horse......" to "......this type of Polish horse......"

Line 66 corrected

In 77, change "....... all the lines' representatives demonstrate breeding acitvity" to " stallions from all of the paternal lines play active roles in breeding activities."

Line 80 corrected

In Line 77-82, Please only keep the latest report on the population sizes of paternal lines, and the previous survey of Pasicka and Tomczyk-Wrona should be deleted. Please add a comma before " nowadays".

Line 80-85. We corrected in that way to save continuity of reports.

In Line 147, change " ......may be treated ......" to " ......and may be treated ......". Again, the authors should provide the results of analysis on genetic distance to support that all of the samples could be merged and analyzed together.

Line 152, there is and before brackets; genetic distance added.

In Line 154-156, the methods should be assigned into Materials and Methods part.

We have other content in those lines (table), please define what do you mean.

Methods are assigned into Materials and Methods.

In Line 169-170, the point" it is caused by low genetic distance among six lineages" should be supported by the results of analysis on genetic distance, as mentioned above.

Added

In Line 173 change " .....was calculated for....." to " .....was observed at loci....."

Line 180 corrected

In Line 174-175, the authors stated that the mean number of alleles for all samples was 7.765, and the mean effective number of alleles was 3.854. However, according to the results showed in Table 1, the number of alleles and the mean effective number of alleles calculated for each sub-population (paternal line) are less than the mentioned values (7.765 and 3.854). Given the very low private alleles of each sub-population, how could the authors get the mean values that are greater than all of the individual ones? Please check that carefully.

Line 182 It was mistake in 7.765, we corrected in 7.176 (rewriting error of 7.1765 and reducing).

All we checked. Six lines have different number of individuals, so average number is from average number in sub-population is little less different (not all sub-population have the same variants). So average for all samples without division (7.176 and 3.854) is not the sum of average values when subpopulation division is included (6.157 and 3.680).

In Line 176, change "The F-Statistics results determined for the whole studied group’s STR markers……" to "The F-Statistics results of the whole studied population……"

Line 183 corrected

In Line 183, change " ……for each locus in the data" to " ……for each locus "

Line 190 corrected

In Line 193, Change "The mean expected heterozygosity was established as 0, .7046 and was lower than the observed heterozygosity (0.6519)" to "The mean expected heterozygosity was 0.7046 and lower than the observed heterozygosity (0.6519)".

Line 201 corrected

In Line 195-196, Change "The PIC, Ho and He coefficients (above 0.6) had in general very high values for all but four loci (HTG4, HTG6, HTG7, and LEX3)" to "The values of PIC, Ho and He coefficients (above 0.6) are very high values except HTG4, HTG6, HTG7, and LEX3."

Line 203-204 corrected

In Line 228, "influence" should be changed to "influences".

I’m guessing line 239. “Such a bottleneck event usually influences” – corrected

In Line 243, " but similar that in to Estonian population" should be replaced with "but similar with that of Estonian population"

Line 256 corrected, but it was change of English editor

In Line 255, please change" could be related with other female lines variants...." to " could be related with the genetic variations of other maternal lines...."

Line 268 corrected, but it was change of English editor

In Line 267, "......the maternal lines; this would ......" should be changed to "......the maternal lines, which would ......"

Line 281 corrected, but it was change of English editor

In 279-280", "....despite the small number of animals" should be changed to " .... Despite that the population experienced a bottleneck event of genetic diversity several decades ago."

Line 294 corrected, but it was change of English editor

Reviewer 2 Report

General: With the exception of a stretch in the discussion, the authors now make no claims that are unsupported by the data. While the language could still be improved, I had no trouble understanding.

Generally, a result of the authors' study is that inbreeding within the breed is negligible, which means that the fixation statistics (F_st, F_is, F_it) are close to zero and negligible. I suggest to re-write the sentence in line 225: "Such a small FIS value is also observed in Thoroughbred [31] and Czech Haflinger horses [32]."

The toublesome paragraph starts then on line 227. I suggest to only use the first two sentences (I edited slightly): "The Polish Konik population was rebuilt from few surviving individuals after 227 World War II. Such a bottleneck event usually lowers genetic diversity." Then I suggest to continue with the next paragraph (also slightly edited): "Nevertheless, compared to other breeds the allelic polymorphism was high in the core panel and slightly lower in the additional panel. We detected..."

To compensate for the omission of these two sentences, the authors could add after the sentence ending in line 265: "Compared to these breeds, our results did not show inbreeding."

To lines 266 and following: Since the breed is maintained at basically random mating, it is highly likely that maternal lines, just like the paternal lines studied in this article, are also not differentiated. It would be more useful to compare data from the same panel of loci in samples of other breeds.

Author Response

General: With the exception of a stretch in the discussion, the authors now make no claims that are unsupported by the data. While the language could still be improved, I had no trouble understanding.

Thank you for all comments and suggestion.

Manuscript has undergone English language editing by MDP before I sent it to you.

Generally, a result of the authors' study is that inbreeding within the breed is negligible, which means that the fixation statistics (F_st, F_is, F_it) are close to zero and negligible. I suggest to re-write the sentence in line 225: "Such a small FIS value is also observed in Thoroughbred [31] and Czech Haflinger horses [32]."

I don’t have in line 225 this sentence. I’m guessing which one do you mean and it was improved so far: (line 235) “The FIS value is similar to those reported for Thoroughbred [31] or Czech Haflinger horses [32] and could be interpreted as no heterozygosity reduction in the population.”

The toublesome paragraph starts then on line 227. I suggest to only use the first two sentences (I edited slightly): "The Polish Konik population was rebuilt from few surviving individuals after 227 World War II. Such a bottleneck event usually lowers genetic diversity." Then I suggest to continue with the next paragraph (also slightly edited): "Nevertheless, compared to other breeds the allelic polymorphism was high in the core panel and slightly lower in the additional panel. We detected..."

We corrected.

To compensate for the omission of these two sentences, the authors could add after the sentence ending in line 265: "Compared to these breeds, our results did not show inbreeding."

I’m sorry, because of mess with numeration of Lines, I don’t know, where I should add it.

To lines 266 and following: Since the breed is maintained at basically random mating, it is highly likely that maternal lines, just like the paternal lines studied in this article, are also not differentiated. It would be more useful to compare data from the same panel of loci in samples of other breeds.

We will for sure. But we also making research on the male horses of the paternal lines with Y-chromosome and female with mtDNA, but in other research (in progress). It is hard to find in literature how many lines there is now, so our goal was point that without masking this breed by another breeds. We have also in progress research of Polish Konik vs. other breeds on STR and SNP.

Reviewer 4 Report

This revised version of the manuscript has a much better flow in the use of English.

However I suggest you revisit Line 212 to 214 "Final and recommended...." as this sentence makes little sense.

You probably mean to say something along these lines:-

" To date there is no recommended panel of SNPs for equine parentage testing. The ISAG panel of STRs is universally accepted and can provide information on population diversity and genetic structure."

I agree with your findings that this study reveals little or no substructure associated with paternal lineage.

I strongly suspect this lack of associated substructure is the reason many of your Fis figures are negative. As such you should not rely on them as justification that there is little or no inbreeding in the population.

Your Mean Fit figures (discounting LEX3 for valid reasons) are much more indicative of the level of inbreeding in your sampled set. This is at 7% which is consistent with the findings of previous researchers and at a level expected in an equine breed with a relatively shallow (7 generation) pedigree.

Author Response

This revised version of the manuscript has a much better flow in the use of English.

Thank you for all comments and suggestion.

Manuscript has undergone English language editing by MDP before I sent it to you.

However I suggest you revisit Line 212 to 214 "Final and recommended...." as this sentence makes little sense.

Sentence was suggested by other Review.

You probably mean to say something along these lines:-

" To date there is no recommended panel of SNPs for equine parentage testing. The ISAG panel of STRs is universally accepted and can provide information on population diversity and genetic structure."

Yes, thank you, it is much better.

I agree with your findings that this study reveals little or no substructure associated with paternal lineage.

I strongly suspect this lack of associated substructure is the reason many of your Fis figures are negative. As such you should not rely on them as justification that there is little or no inbreeding in the population.

Your Mean Fit figures (discounting LEX3 for valid reasons) are much more indicative of the level of inbreeding in your sampled set. This is at 7% which is consistent with the findings of previous researchers and at a level expected in an equine breed with a relatively shallow (7 generation) pedigree.

We agree. But we didn’t find any other, better justification supported by proofs. We believe that maternal testing will give a broader context. And our next research (male Y-chromosome and female with mtDNA, and also SNP) will explain much more.

This manuscript is a resubmission of an earlier submission. The following is a list of the peer review reports and author responses from that submission.

Round 1

Reviewer 1 Report

In the study, the authors revealed the genetic diversity of Polish Konik horses with 17 satellite DNA loci, by sampling 196 samples from six paternal lineages. The results showed that there was little genetic difference among the six paternal lineages, and no significant inbreeding in the studied population, and the majority of the genetic variation was from the difference among individuals. The results are valuable for assessing the genetic diversity of Polish Konik horses and conserving the breed.

Though the authors provided the detailed information of the Characteristics of 17 microsatellite loci, genetic structure, F-statistics and Polymorphic analysis of the population, but more comprehensive analysis should conducted with the original data. For example, DA genetic distance should be analyzed to confirm the authors' conclusion that there is no significant genetic difference between the six paternal lineages. And ideally, a dendogram of phylogenetic relationships among the 6 populations using neighbor joining method and DA genetic distance should be provided by the authors.

Polish Konik horses had experience bottleneck events of genetic diversity (The size of the population had decreased to limited numbers in the recent history), so it is expected that there is some degree of inbreeding (if not high) in the population. But there is no inbreeding found according to the authors' results. Authors' should discuss it and give a reasonable explanation in the discussion part.  

As a study aimed to the conservation of Polish Konik horses, the authors should explain: what do the data and results mean to the conservation of the horse population? Are they in a good state of conservation? What are the main problems faced referring to the results of the present study? Suggestions should be given about how to resolve the problems.

Other problems:

1. The authors should provide more detailed sample information, including the sample sizes of the six paternal lineages, how many male horses sampled in each line, and the locations of the samples.

2. The means (average values) of the F-statistics (FIT, FST and FIS) should be provided in the Table 2.

3. The authors stated that animals origins and differences among individuals accounted for about 3% and 97% of genetic variation, respectively (in abstract, Line 17-18; in Discussion, Line 192-193). However, the result was not showed in the Results part. How did the authors obtain it?The authors should add the detailed information in the Results part, and also provide the method (which was used to estimate the proportion of genetic variation) in the Materials and Methods part.  

4. The English writing of the manuscript should be improved, ideally to be checked by a native English speaker.

Reviewer 2 Report

The authors present a population genetic study of Konik horses, a breed kept in semi-feral conditions. The data originate from 17 autosomal microsatellite loci, which are used for parentage control. As six male lines are known for this breed, the authors are interested in the genetic differentiation among these lines, especially since the current representation of the male lines in the population is uneven. The breed as a whole, however, seems to be managed as a panmictic population (as is evident from the results). Hence it is not surprising that nearly no differentiation within the Konik population is found with the autosomal loci, neither when information from the paternal line is used, nor when the program "Structure" is used to find cryptic population differentiation. Since the population is nearly panmictic, it makes little sense to preserve genetic variability by trying to balance the proportion of male lines. Such a breeding program would inevitably lead to loss of genetic variation not associated with the male lines (which is the whole autosomal and the mitochondrial genome). 

The most positive finding of the study is a parental exclusion probability with these microsat loci of nearly 100%. Obviously this would not be possible, would the breed be strongly inbred. Nevertheless, the claim that there was "no inbreeding in Polish Konik population for studied population" is not supported by the data. (This verbatim quote shows some problems with the English language, which did not distract too much from the content.)

When estimating inbreeding, the reference population is important. It makes no sense to estimate the inbreeding coefficient within a single panmictic population, which the Konik population seems to be. Rather other populations or breeds would be needed as reference. As these microsatellites seem to be used for parentage control in other breeds too, it should be easy to obtain such data. Then determination of inbreeding relative to a base horse population (at about the time of the establishment of modern breeds) would be possible. Without such information, only the coarse comparison of heterozygosities of the loci relative to those in other similar studies is possible (as in the discussion). This comparison is imprecise, as the expected heterozygosity depends on the sampling design wrt to both individuals and loci. The comparison seems to show similar or slightly lower heterozygosity than similar semi-feral breeds. These other breeds may, however, show inbreeding themselves with respect to a base horse population (see above).

In summary, the authors seem confused about the role of the base population when estimating inbreeding. With data from the same loci but different breeds, F_st relative to other breeds and thus population level inbreeding could be measured. Such data may easily be available.

Reviewer 3 Report

In this paper, authors describe the genetic diversity and population structure of six paternal lines belonging to the Polish Konik horse. The analyses revealed that there was no inbreeding in the studied population, the genetic diversity was similar to other horse breeds from different countries and the Polish Konik horse should be monitored to preserve its genetic variability. This study is interesting and results improve our knowledge on Polish Konik genetic diversity and structure.

I have some suggestions for the presentation of the study.

Simple Summary

L12: remove “In this study”, it is not needed.

L12: replace with “We investigated the genetic diversity of Polish Konik horse in sire lines.”

L13: replace “The Polish Konik horse” with “It”.

L18: what do you mean with “animals origin”? Historical? Genetic? Please clarify.

L21-22: authors should explain in a better way the statement “for making informed decisions in paternal lineages management”.

Abstract

L23: remove “(Equus caballus)”. It is unnecessary in the abstract.

L26-28: there is a repetition.

L30: remove “genetic “ before “structure”.

L32: authors should replace “low” (for Fst) with a value.

Introduction

The introduction focuses on the history and the breeding conditions of Polish Konik horse, important to contextualize the breed, but I think it lacks the state of art about the informative role of microsatellite markers in studying structure and genetic diversity in this and other horse breeds.

Moreover, authors should highlight the aim of study at the end of the Introduction.

L52: replace “recommended nomenclature by” with “nomenclature recommended by”.

L54: what individuals were placed in the oldest National Polish Stud? Stallions or mares? And from which breed?

L58-59: authors should add the verb (probably “was established”) at the of the sentence “….the first volume of the studbook for the breed”.

L71: replace “active is only 16” with “only 16 are active”.

L75-77: change the sentence with “All male lines (…) demonstrate a sufficient breeding activity and the least……”.

L80: replace “maintenance” with “preservation”.

L84: replace “the genetic variability of Polish Konik” with “the genetic variability of this breed”.

Materials and Methods

I think that a table (as supplementary) could be useful to know the characteristics (geographic origin, pedigree data, etc.) of each sample.

L94: replace “represented” with “representing”.

L99: authors should delete “-” after “protocols”.

L106-107: authors should substitute “:” with parentheses.

L110: F statistics.

L119-120: authors claim to have used “our own software”. Which software? Please clarify and describe it.

Results

L140: authors should add a title for Figure 1.

L145: what do you mean with “particular”?

L148: remove “there”.

L152-153: please write a more fluent sentence.

L159: please add “that” after “suggested”.

L170: replace “despite” with “except for”.

Discussion

L176: add “-” between “areas” and “were”.

L183: remove comma after “proven”.

L183: replace “analysed” with “analysis”.

L184: replace “differentiated and explained” with “differentiate and explain”.

L185: “of many horse breed”.

L185: remove “the” before mtDNA and replace “mtDNA” with “mitochondrial DNA”, as is present throughout the text only once.

L187: “paternal”.

L189: “as a single population”.

L190: add the article “the” before “number”.

L195: delete “the” before “inbreeding”.

L196: replace “similar to the reported” with “similar to that reported”.

L200: “these loci” or “this locus”?

L212: “as in Gralak et al.”, delete “and”.

L214: replace with “…Jaca Navarra, two horse breeds included in a conservation program….”.

L217: replace “points” with “point”.

L218: replace with “…of applied marker set for a population study in the investigated breed”.

L221: replace “what” with “that”.

L221: replace comma between “comprehensive” and “intentional” with “and”.

Conclusions

L229: replace “in Polish Konik population for analysed population” with “in the analysed Polish Konik population”.

L229: replace “Despite that” with “Despite the”.

L230: replace with “similar to that obtained”.

L232: the statement “now particular paternal lineages have uneven representation” should be better explained.

In conclusion I consider this a paper providing new information and suggest Animals to consider it for publication, after revision.

Reviewer 4 Report

1) Abstract, Lines 26 – 28: Replace two similar sentences by a sentence that is better formulated.

2) Introduction, Lines 58 – 59: Need to make the sentence starting “In 1955 the Polish register…” better (missed verb in the passive voice?).

3) Introduction, Lines 78 – 79: need to change the order of words in “conservation program breeding” to “conservation breeding program”.

4) Materials and Methods, Line 112: Correct “Marcov” to Markov.

5) Materials and Methods, Lines 114 – 117: Mark a period after [21]. Begin a new sentence with “However, because of …”.  Move the numbers of individuals in paternal lineages from here to the end of the 1st paragraph of Materials and Methods (Line  94).  

6) Materials and Methods, Lines 117 – 125: Make one paragraph instead of two.

7) Table 1, Lines 129 – 131: The authors defined “Na” as “No. of alleles”. I think that need to replace “No. of alleles” with “mean number of alleles”.

I do not recommend using “Ne” for “No. of effective alleles” because Ne is an acronym usually used for the effective population size. It is better to replace “Ne” with Ae, as the effective number of alleles.   

The authors need to replace “No. private alleles” by the mean number of private alleles or the frequency of private alleles. I am not sure what they show in that column.

Replace “>=” with ≥. What is “freq.”? I understand it is a frequency, but they need to include a full term.

The authors need to add the table with Ho, observed heterozygosity.

I would also recommend including a sample size (N) for each paternal lineage. It is better to replace “Population” by “Paternal lineage”.

8) Results, Line 128: “in general terms may be treated as a single population”. What “general terms” do you mean? Please explain your statement based on the data in Table 1.

9) In Results, need to clearly describe the structure analysis results. Improve the sentence “Analysis of the structure suggested…” (Lines 148 – 150) and place it after “… presented at Figure 2.” in Line 138.

10) Results, Lines 145 – 146: Make two sentences.

11) Results, Lines 150 – 151: Do you mean “mean effective number of alleles”?

12) Results, Lines 152 – 153: Need to improve both sentences.

13) Results, Lines 152, 153, 158 – 160: These results need to be summarized in one paragraph. Make two sentences from “Mean Fit was low…” (Lines 158 – 160).

14) Table 2, Lines 154 – 157. Include the mean values.

15) Need to make a conclusion, based on the data, if you found the deviation from Hardy-Weinberg equilibrium in the autosomal STR loci.

16) Table 3, Line 162: Replace “Polymorphic” by “Population”.

17) Discussion, Line 175: Replace “  ̶  like” by “such as”.

18) Discussion, Lines 177 – 178: I recommend to delete “for example”.

19) Discussion, Line 181: What is “had influence”? Please, be specific. Replace “for example” with “of”.

20) Discussion, Lines 183 – 186: The authors need to follow the logic of their first sentence in the paragraph. If “Y chromosome variability” explained “the origin of male founders” they need to say something about mtDNA and the origin of female founders. Separate mtDNA and STRs you tried to join in the second sentence of the paragraph. Correct the first sentence in the paragraph: “can” is used two times, “can differentiated and explained” need to be corrected.

21) Discussion, Line 187: correct “patrnal” lines.

22) Discussion, Line 190: Show the number of genetic groups based on STRs and explain.

23) Line 191 vs Line 230: Correct the discrepancies “the small diversity of analyzed loci” vs “genetic diversity was high”.

24) Discussion, Line 192: Explain “non-random mode of sample selection”; cannot find the explanation in Materials and Methods.

25) Discussion, Lines 192 – 194: I would recommend moving the sentence “About 3% (0.0305) of …” to Results.

26) Discussion: The authors need to provide a systematic and clear comparison of their results and statistics with other breeds. They just mentioned a few breeds without a clear comparison.

27) Discussion, Lines 201, 206: Use a different order for the references; instead “Mackowski et al. [33] and Gralak et al. [1]”, the order “Gralak et al. [1] and Mackowski et al. [33]” looks better.

28) Discussion, Lines 198 – 207: What is difference between your group of horses and the groups of the Polish Konik horses used in [1] and [33]?

29) Discussion, Line 208 and References, Lines 320 – 322: Your reference [34] is incorrect. Probably, you mean the article entitled “Population genetic analysis of the Estonian native horse suggests diverse and distinct genetics, ancient origin and contribution from unique patrilines”. Please, correct pages as well.

30) Discussion, Lines 208 – 210: Castaneda et al., 2019 provided the detailed analysis of the Polish Konik breed. Please, discuss and compare your results and the results published by Castaneda et al., 2019.

31) Discussion, Lines 211 – 223: the paragraph is unclear. Provide a clear comparison of the statistics with other breeds.

32) Conclusions, Lines 225 – 234: What is the novelty of your study in the comparison with other studies of the Polish Konik breed? What is the significance of your study and results? The authors did not provide a strong message on the significance and importance of their results.

33) Conclusions, Lines 228 – 229: the conclusion on the lack of inbreeding does not have a clear discussion in Results and Discussion.
